# Adhesion pilus retraction powers twitching motility in the thermoacidophilic crenarchaeon *Sulfolobus acidocaldarius*

Arthur Charles-Orszag [1], Marleen van Wolferen [2], Samuel J. Lord [1], Sonja-Verena Albers [2,3] ✉ & R. Dyche Mullins [1] ✉

Type IV pili are filamentous appendages found in most bacteria and archaea, where they can support functions such as surface adhesion, DNA uptake, aggregation, and motility. In most bacteria, PilT-family ATPases disassemble adhesion pili, causing them to rapidly retract and produce twitching motility, important for surface colonization. As archaea do not possess PilT homologs, it was thought that archaeal pili cannot retract and that archaea do not exhibit twitching motility. Here, we use live-cell imaging, automated cell tracking, fluorescence imaging, and genetic manipulation to show that the hyperthermophilic archaeon *Sulfolobus acidocaldarius* exhibits twitching motility, driven by retractable adhesion (Aap) pili, under physiologically relevant conditions (75 °C, pH 2). Aap pili are thus capable of retraction in the absence of a PilT homolog, suggesting that the ancestral type IV pili in the last universal common ancestor (LUCA) were capable of retraction.

Type IV pili are polymeric fibers that project from the outer surface of most archaea and bacteria[1–3] where they function as an important interface between these cells and their environment. For example, type IV pili mediate adhesion to biotic and abiotic surfaces, inter- and intraspecies cell-cell interactions, biofilm formation, cell motility, and uptake of DNA in naturally competent species[2,4–8]. Type IV pili are also receptors for bacteriophages and archaeal viruses[9–13], as well as central virulence factors in a number of plant and human bacterial pathogens[14]. Despite decades of work, however, the type IV pili of archaea remain much less understood than their bacterial counterparts.

Similar to their bacterial counterparts, archaeal type IV pili are assembled from one or two major pilin proteins that polymerize into filaments. Prepilins contain a signature N-terminal sequence (the class III signal peptide) and are processed to mature pilins by the prepilin peptidase, PibD[15,16]. Type IV pilus assembly machinery in archaea appear to comprise fewer components than those of bacteria. Typically, operons encoding the assembly factors contain genes for: (i) the assembly platform protein and (ii) the secretion/assembly ATPase[1].

Depending on the type of pilus formed, additional genes may be present that encode proteins of unknown function[1,17,18]. Finally, phylogenetic analysis revealed that genes encoding specific S-layer proteins which might be important for the integration of the pilus into the archaeal cell envelope[19] are present in close proximity to type IV pili genes clusters.

In most bacterial type IV pili, the antagonist actions of two conserved ATPases, PilB and PilT, form the basis of pili dynamics. While PilB promotes fiber assembly, ATP hydrolysis by PilT fuels pilus disassembly and retraction. PilT-mediated retraction of a single pilus can generate forces of ~100 pN and drive retraction at speeds of up to 1 μm per second[20,21]. In bacteria, this dynamic behavior of type IV pili generates characteristic saltatory motions of surface-adhered cells, called "twitching" motility. In some rod-shaped bacteria twitching is highly directional and coupled to chemo-, mechano- or phototaxis[22–24], while in other non-rod-shaped species[5,25] it appears to be random. Twitching motility is essential to surface colonization, biofilm formation, and dynamic self-aggregation[5,26,27]. For many years PilT was thought to be essential for retraction of type IV pili, leading to the proposal that

[1]Department of Cellular and Molecular Pharmacology, Howard Hughes Medical Institute, University of California, San Francisco, San Francisco, CA, US. [2]Molecular Biology of Archaea, Faculty of Biology, Institute of Biology II, University of Freiburg, Freiburg, Germany. [3]Signalling Research Centre BIOSS and CIBBS, Faculty of Biology, University of Freiburg, Freiburg, Germany. ✉e-mail: sonja.albers@biologie.uni-freiburg.de; dyche.mullins@ucsf.edu

archaeal pili are not capable of retraction and do not employ twitching motility[3,28,29]. Recent work, however, revealed that some bacteria exhibit PilT-independent pilus retraction[30–32]. While performing live-cell, time-lapse imaging of the model hyperthermophilic crenarchaeon *Sulfolobus acidocaldarius* at their normal growth temperature (75 °C) we previously observed a type of surface-motility reminiscent of bacterial twitching motility[33], challenging the idea that type IV pili in these cells are unable to retract.

The genome of *S. acidocaldarius* encodes three different type IV pilus systems that are well studied. The first system creates the archaellum, a motile pilus assembled via a type IV pilus assembly machinery that −in contrast to other type IV pili− rotates to generate propulsion for swimming motility[34,35]. The second system creates a UV-inducible pilus (Ups), which initiate species-specific, cell-cell contacts that enable DNA exchange and DNA repair in Sulfolobales[4,6]. The third system assembles an adhesive structure called Aap pilus[17]. This pilus is important for adhesion to surfaces and formation of biofilms[18] and can be targeted by viruses as a cell adhesion factor[10–13].

In this work, we describe the surface motility of wildtype *S. acidocaldarius* cells and several mutant strains lacking one or more cell surface appendages. Automated cell tracking in live-cell imaging assays at high temperatures revealed that cells are capable of bona fide twitching motility, and that adhesion pili are specifically responsible for this phenomenon. Moreover, using super-resolution, live-cell, fluorescence imaging at high temperature, we directly observed the rapid retraction of adhesion pili, and the coupling of these retraction events to saltatory cell movements. Finally, to determine the generality of these results, we analyzed cell motility in other Sulfolobales species and discovered that twitching motility is not unique to *S. acidocaldarius*. Our results indicate that PilT-independent pilus retraction might be common in archaea and strongly suggest that the ancestor of type IV pili in the last universal common ancestor (LUCA) was already capable of retraction and may have powered twitching motility.

## Results

### Adhesion pili mediate twitching motility in *S. acidocaldarius*

To study the motility of *S. acidocaldarius*, we transferred cells from exponentially growing cultures to an environmentally controlled microscope chamber heated to 75 °C and performed timelapse imaging using Differential Interference Contrast (DIC) microscopy. In these experiments, we imaged microscope fields containing multiple cells for five minutes at one frame per second. We used automated segmentation and tracking with a trainable detector to identify and follow the movement of multiple cells in each timelapse movie. This machine learning approach enabled us to track several cells per field over hundreds of frames. The resulting tracks were then analyzed (Fig. 1).

As we previously observed for wild-type *S. acidocaldarius*[33], glass-adhered cells from the parental strain (MW001, herein "WT") were motile. Surface motility in WT cells appeared to be a saltatory random walk (Fig. 1a, d, and Supplementary Movie 1). The average track displacement over 5 min was $5.7 \pm 0.44\,\mu m$. Not all cells were motile, however, with 12.4% of tracks having a total displacement of less than $2\,\mu m$ (the observed mean cell diameter was $1.6\,\mu m$ under these experimental conditions[33]) (Fig. 1g, h). WT tracks showed an average persistence ratio of $0.12 \pm 0.03$, showing low directionality (Fig. 1i). For comparison, primary rat astrocytes exhibit a persistence ratio of 0.75 in a wound healing assay[36]. To better understand the nature of this motility, we created log-log plots of the mean square displacement (MSD) of WT trajectories versus time. The slopes of these log-log MSD plots cluster just below a value of 1.0 with very few slopes above 1.2 (Fig. 1j, k). These values are consistent with a random walk and display no evidence of directed motility. Interestingly, MSD analysis also revealed a smaller population of WT trajectories with slopes much less

than 1.0, suggesting strongly subdiffusive motion, possibly by constrained or tethered cells (Fig. 1k).

To determine which type IV pilus subtypes are involved in twitching motility, we analyzed surface motility in deletion strains (Table 1) missing either the archaellum platform protein ArlJ (Δ*arlJ*), the UV-inducible pili assembly ATPase UpsE (Δ*upsE*), or both (Δ*upsE*Δ*arlJ*). All of these mutant strains exhibited motility indistinguishable from wildtype cells, with average total displacements of $4.6–6.6\,\mu m$ (Fig. 1c, f, g, h, Supplementary Fig. 1 and Supplementary Movie 1) and persistence ratios of $0.11–0.12$ (Fig. 1i). Interestingly, while Δ*upsE* and Δ*upsE*Δ*arlJ* had a proportion of non-motile cells comparable to that of the WT (18.4% and 14.7% respectively) nearly all Δ*arlJ* cells were motile, with only 4.2% of tracks having a displacement below $2\,\mu m$ (Fig. 1g, h, Supplementary Fig. 1 and Supplementary Movie 1). Interestingly, MSD analysis showed that Δ*arlJ* trajectories exhibit mainly diffusive twitching behavior, with a loss of the subdiffusive or tethered population (Fig. 1k). In sharp contrast, deleting the adhesion pilus assembly protein AapF (Δ*aapF*) abolished twitching motility, with surface-adhered cells mainly wobbling around a fixed point (Fig. 1b, e). Accordingly, track analysis showed that 46.2% of Δ*aapF* cells had a total displacement below $2\,\mu m$ (Fig. 1h). Average track displacement ($1.49\,\mu m$) and persistence ratio ($0.02$) in Δ*aapF* cells were both significantly smaller than in WT cells (Fig. 1g, h), and MSD analysis confirmed that Δ*aapF* trajectories were mostly subdiffusive, or tethered, indicating that Aap-deficient cells are non-motile (Fig. 1j, k).

In our experiments *S. acidocaldarius* exhibits bona fide twitching motility as seen in type IV pili-expressing bacteria. Moreover, a strain expressing only Aap pili (Δ*upsE*Δ*arlJ*) exhibits motility comparable to that of WT cells, while a strain deficient in Aap pilus assembly (Δ*aapF*) is non-motile. The adhesion pili (Aap) are thus necessary and sufficient to drive twitching motility in *S. acidocaldarius*, while archaella and UV-inducible pili appear not to be involved in this process.

### Twitching motility in *S. acidocaldarius* is powered by retractile adhesion pili

Since archaea lack a PilT-like retraction ATPase[2,29,37], we sought to visualize type IV pilus dynamics in live *S. acidocaldarius* cells to determine whether surface motility is associated with pilus retraction. To image cells and their pili, we non-specifically labeled surface-exposed lysines of membrane-associated proteins with an AlexaFluor dye functionalized with an N-Hydroxysuccinimide (NHS)-ester, and imaged the labeled cells at high-temperature (Fig. 2). We used Δ*upsE*Δ*arlJ* cells as they assemble only Aap pili and no other type IV pilus structures. Transmission electron microscopy of negatively stained cells confirmed the presence of Aap pili with similar numbers and lengths to those of WT cells (Supplementary Fig. 3). Labeled cells adhered to the glass surface and underwent twitching motility, indicating that Aap pili were functional. In addition to a bright fluorescent signal originating from cell wall proteins, we detected Aap pili as thin, surface-attached fibers of up to $14\,\mu m$ long. Most importantly, cell movement coincided with rapid retraction of Aap pili in two ways: cells either moved forward in the direction of pilus retraction or moved backward upon loss of surface attachment after a pilus retracted (Fig. 2a, c and Supplementary Movies 2 and 4). The retraction of Aap pili also supported dynamic direct cell-cell interactions (Fig. 2b and Supplementary Movie 3). In cells where adhesion was mediated by a single adhesion pilus, the average pilus retraction speed was $0.3\,\mu m.sec^{-1}$ (Fig. 2c, d and Supplementary Movie 4). The speed of retraction did not correlate to pilus length (Fig. 2e). Similar type IV pilus dynamics were observed in WT cells, which, in these growth conditions, are only expected to express Aap pili (Supplementary Fig. 2 and Supplementary Movie 5). However, the average pilus retraction speed was $0.3\,\mu m.sec^{-1}$.

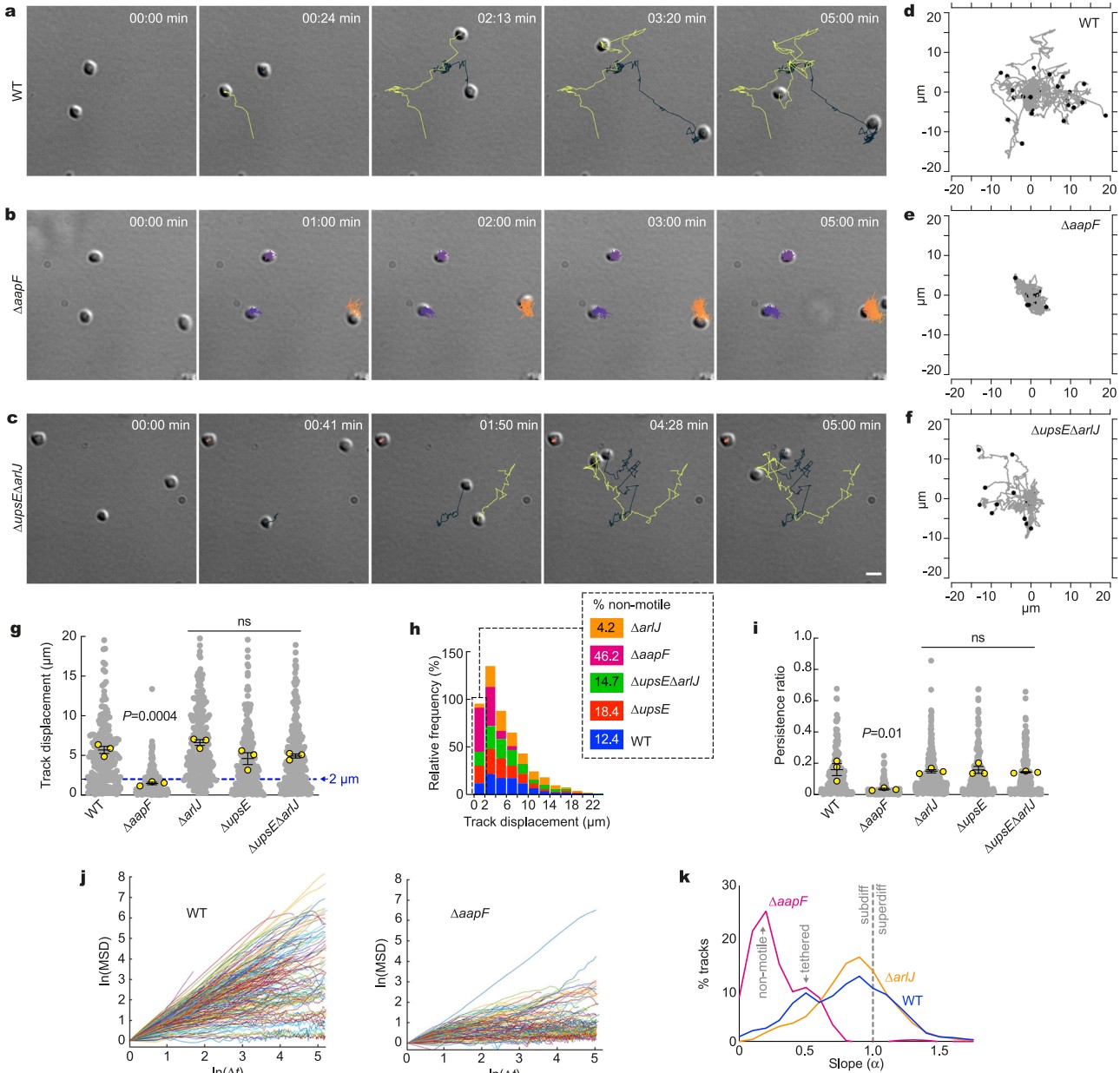

**Fig. 1 | Automated tracking of type IV pili mutants in *S. acidocaldarius* at high temperature. a–c** Differential interference contrast (DIC) live-cell imaging of indicated *S. acidocaldarius* strains at 75 °C. Shown are selected stills representing noticeable steps of glass-adhered cells over a five-minute observation window. Overlaid tracks were automatically obtained with TrackMate 7 after machine-trained detection of the cells with Weka. **d–f** Worm plots of 15–30 representative tracks for each indicated strain. **g** Track displacement measured over five minutes for each indicated strain. **h** Distribution of individual track displacements in the indicated strains. Tracks with a displacement under 2 µm correspond to non-motile cells. **i** Track persistence ratio in the indicated strains. Gray scatter dot plots

correspond to the total number of cells analyzed (N = 200 WT, 222 Δ*aapF*, 284 Δ*arlJ*, 171 Δ*upsE*, 245 Δ*upsE*Δ*arlJ*). In **g** and **i**, superimposed yellow circles represent average values from each of n = 3 independent experiments and error bars represent the mean ± SEM of those three biological replicates. *P* values were calculated on the replicate means using a one-way ANOVA test with a Tukey's correction for multiple comparisons with a single pooled variance. **j** Mean square displacement (MSD) analysis of all tracks in WT and Δ*aapF* strains. **k** Distribution of slope value α in MSD analysis of all tracks in WT, Δ*aapF* and Δ*arlJ* strains. Source data are provided as a Source Data file.

These results demonstrate that *S. acidocaldarius* Aap pili are capable of retraction, and that Aap pilus retraction powers twitching motility in these cells.

### Role of the minor pilin AapA
To investigate the molecular mechanisms of Aap pilus retraction in more detail, we investigated twitching in *S. acidocaldarius* strains deficient for other genes in the *aap* operon (Fig. 3). In addition to the major pilin AapB, the ATPase AapE, and the membrane protein AapF,

the *aap* pilus operon comprises two other genes, *aapA* and *aapX*, whose functions are unclear (Fig. 3a). AapX is annotated as an iron-sulfur oxidoreductase, and deletion of the *aapX* gene leads to a lack of pilus assembly[17]. AapA is a type IV pilin that contains a class III signal peptide that is processed by the prepilin peptidase PibD, and it was proposed to be a minor pilin that might co-assemble in the pilus fiber with AapB. AapA was first thought to be indispensable for pilus assembly[17], but a recent study demonstrated that a Δ*aapA* mutant still assembles pili[38], suggesting that AapA might not co-polymerize with

**Table 1 | List of strains used in this study**

| Strain | Description | Source |
|---|---|---|
| MW001 | Δ*pyrEF* (91–412 bp); parental strain | Ref. 57 |
| MW109 | Δ*upsE* (deletion of *saci1494* in MW001) | Ref. 57 |
| MW153 | Δ*aapA* (deletion of *saci1174* in MW001) | Ref. 17 |
| MW154 | Δ*aapB* (deletion of *saci1179* in MW001) | Ref. 17 |
| MW155 | Δ*arlJ* (formerly *flaJ*, deletion of *saci1172* in MW001) | Ref. 18 |
| MW156 | Δ*aapF* (deletion of *saci2318* in MW001) | Ref. 18 |
| MW158 | Δ*arlJ* Δ*upsE* | Ref. 18 |
| MW159 | Δ*aapX* (deletion of *saci2316* in MW001) | Ref. 17 |
| MW160 | Δ*aapE* (deletion of *saci2317* in MW001) | Ref. 17 |
| *Saccharolobus solfataricus* P2 | | DSMZ, gift from Rachel Whitaker |
| *Sulfolobus islandicus* REY15A | | Gift from Quinxin She, gift from Rachel Whitaker |
| *Sulfolobus islandicus* M16.4 | | Gift from Rachel Whitaker |

AapB in the pilus fiber, and that AapA does not have a function in pilus assembly. Transmission electron microscopy of negatively stained cells confirmed that Δ*aapA* was piliated (Supplementary Fig. 3).

Surprisingly, the Δ*aapA*, Δ*aapB*, or Δ*aapX* deletion mutants all exhibited drastically reduced twitching motility in our live-cell imaging assay (Fig. 3 and Supplementary Movie 6). Similar to the Δ*aapF* mutant cells (Fig. 1), most cells were non-motile (Fig. 3b, c and d). However, in Δ*aapB*, and to a lesser extent in Δ*aapX*, some cells appeared very loosely attached to the surface. As a result, these cells would detach and roll away over long distances because of the high convection flows in the chamber (Fig. 3b), generating long tracks (Fig. 3c) with a higher persistence ratio (Fig. 3e). These data are consistent with previous literature showing that Δ*aapB* and Δ*aapX* do not assemble pili. Although the putative minor pilin AapA is not required for pilus formation, and we observed pilus retraction in a minor subset of Δ*aapA* cells (Supplementary Fig. 2 and Supplementary Movie 7) we nevertheless found that this pilin does appear to play an important role in Aap-pilus mediated twitching motility. Understanding the nature of this role will require additional studies.

**Twitching motility is present in other Sulfolobales**

Other members of the Sulfolobales encode an Aap pilus operon[17]. To test whether type IV pilus-mediated twitching motility is specific to *S. acidocaldarius*, we investigated cell motility in other Sulfolobales species that express adhesion pili. Automated tracking of cells imaged at high temperature showed that *Sa. solfataricus* and *S. islandicus* M16.4 and REY15A all include some cells that exhibited twitching motility, although with a lower average track displacement and with a higher proportion of non-motile cells overall compared to *S. acidocaldarius* (Fig. 4 and Supplementary Movie 8). Attempts to visualize type IV pilus dynamics using the labeling strategy that worked for *S. acidocaldarius* failed in *S. islandicus*. We did, however, detect type IV pilus retraction events in *Sa. solfataricus* associated with surface motility and dynamic cell-cell interactions similar to those observed in *S. acidocaldarius* (Supplementary Movies 9 and 10).

Our data strongly suggests that twitching motility is also supported by type IV pili retraction in *Sa. saccharolobus* and *S. islandicus*, and demonstrates that twitching motility is not restricted to *S. acidocaldarius* but common in Sulfolobales. It also indicates that PilT-independent type IV pilus retraction may be widespread among archaea.

## Discussion

At any given moment most prokaryotes on planet Earth are forming part of a biofilm[39,40] often comprising multiple species that either coexist in synergy or compete for resources, making these environments key drivers of evolution. The formation of biofilms is dependent on the ability of cells to adhere to, explore, and eventually colonize surfaces. In both bacteria and archaea, type IV pili are central players of biofilm formation as they promote surface attachment. In most bacteria, the antagonist actions of the conserved ATPases PilB and PilT promote the assembly and retraction of pilus fibers, thereby powering twitching motility, which is essential for surface exploration and dynamic cell-cell interactions. The type IV pilus assembly machinery is ubiquitous in prokaryotes and was likely present in the last universal common ancestor, LUCA[2,19,28]. However, the retraction ATPase, PilT, is only found in bacteria, leading to the assumption that archaea are not capable of twitching motility[3,28,29]. However, recent observations of live *S. acidocaldarius* cells challenged this concept as the cells appeared highly surface-motile, in a manner reminiscent of bacterial twitching motility[33].

In the present work, we show that *S. acidocaldarius* cells deficient for adhesion pili are no longer surface-motile, whereas deletion of any of the other type IV pili subtypes (archaella and UV-inducible pili) did not impair surface motility. Given the saltatory and non-directional mode of motility in WT cells, these results demonstrate that *S. acidocaldarius* uses twitching motility to adhere to and explore surfaces, and that this behavior is specifically mediated by adhesion pili. Of note, our data shows that cells deficient for archaella assembly (Δ*arlJ*) have a higher proportion of motile cells compared to the WT, suggesting the existence of a genetic crosstalk between type IV pilus subtypes. Indeed, it was previously observed that adhesion pili are upregulated in a Δ*arlJ* mutant[18]. Additionally, it is possible that WT cells express a small amount of archaealla in exponential phase that promote surface tethering, as indicated by our MSD analysis of WT cells trajectories compared to Δ*arlJ* cells. Similarly, archaella-mediated tethering may explain why non-piliated mutant Δ*aapF* cells, which are hyperarchaellated[18], display higher surface adhesion (Fig. 1) than Δ*aapB* and Δ*aapX* mutants (Fig. 3), which are not hyperachaellated. Alternatively, other appendages like archaeal threads[41] might contribute to pili-independent surface adhesion.

The fact that *S. acidocaldarius* adhesion pili allow twitching motility implies that adhesion pili are dynamic, either via retraction or via shedding. However, as described earlier, archaea do not express homologs of bacterial PilT ATPases responsible for pilus retraction. Here, super-resolution fluorescence microscopy of living cells clearly shows that cells move around by retracting surface-adhered pili. There, pili appear as thin extended threads that seem adhered through the pilus tip, as proposed for some bacteria[42]. Upon pilus retraction, *Sulfolobus* cells can move in the direction of retraction. In the case where cells are attached to the surface through multiple pili, they move in a direction that brings the cell body in the middle of the remaining adhered pili. These results demonstrate that adhesion pili are retractile.

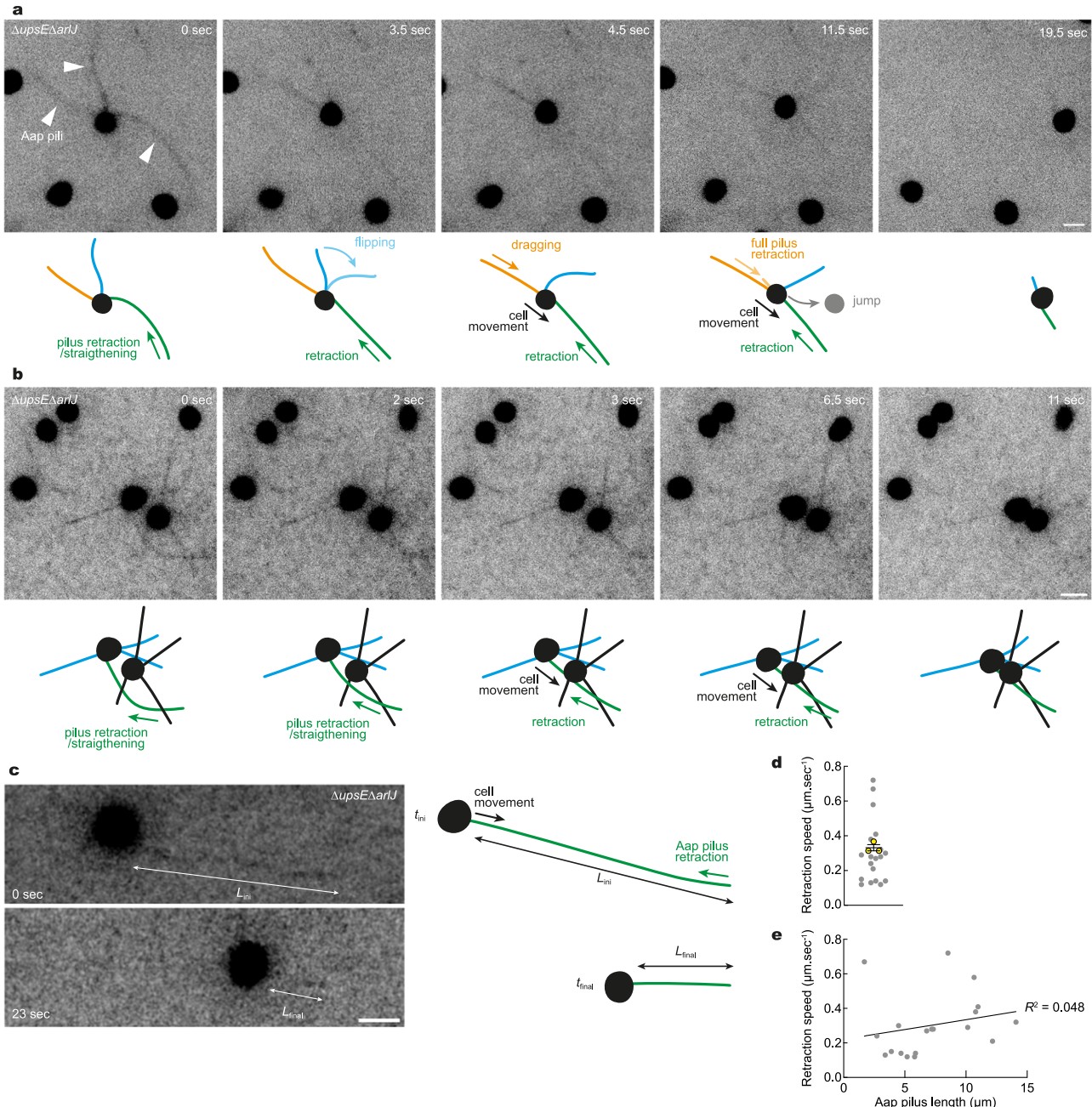

**Fig. 2 | Super-resolution fluorescence imaging of *S. acidocaldarius* adhesion pili dynamics at high temperature.** Surface proteins of *ΔupsEΔarlJ S. acidocaldarius* cells were labeled non-specifically with AlexaFluor 568 NHS-ester. Glass-adhered cells were imaged in Structured Illumination Microscopy (iSIM) every 500 ms. Shown are selected stills from representative movies. **a** Aap pili mediate dynamic adhesion and twitching motility in *S. acidocaldarius*. **b** Pilus retraction mediate dynamic cell-cell interactions in *S. acidocaldarius*. **c** Single Aap pilus retraction events were used to measure pilus retraction speed. Retraction speed is given as the difference between final ($L_{final}$) and initial ($L_{ini}$) pilus length over the difference between $t_{final}$ and $t_{ini}$. **d** Mean retraction speed in single-pilus retraction events in 19 cells. Superimposed yellow circles represent average values from each of $n = 3$ independent experiments. Error bars represent the mean ± SEM of those three biological replicates. **e** Retraction speed is not correlated with pilus length. Scale bars, 2 μm. Source data are provided as a Source Data file.

Our data raise the question of how Aap pili can retract in the absence of a homolog of the PilT retraction ATPase, a problem that is also found in the bacterial world. The bacterium *Caulobacter crescentus* lost a PilT retraction ATPase which was probably present in ancestral species, and relies on the PilB ATPase for both assembly and retraction of the tight adherence (Tad) pilus[30,31]. Interestingly, bacterial Tad pili are thought to have been acquired from archaea through horizontal gene transfer[29]. In the bacterial pathogen *Vibrio cholerae*, PilT promotes both pilus assembly and retraction[32]. Finally, type IV pili

of the bacterial pathogen *Neisseria gonorrhoeae* exhibit slow, ATPase-independent retraction, which can be enhanced by PilT homologs[43].

This motor-independent pilus retraction, however, occurs at speeds around 40 nm.sec⁻¹ and only produces forces in the range of 5 pN, insufficient for twitching motility. In contrast, Aap pili in *Sulfolobus* retract at speeds ranging from 0.3 to 2 μm.sec⁻¹. It was difficult to determine whether the assembly ATPase AapE is involved in pilus retraction as deletion of AapE leads to a lack of assembly of Aap pili[17]. We attempted to overexpress AapE Walker A and B mutants from

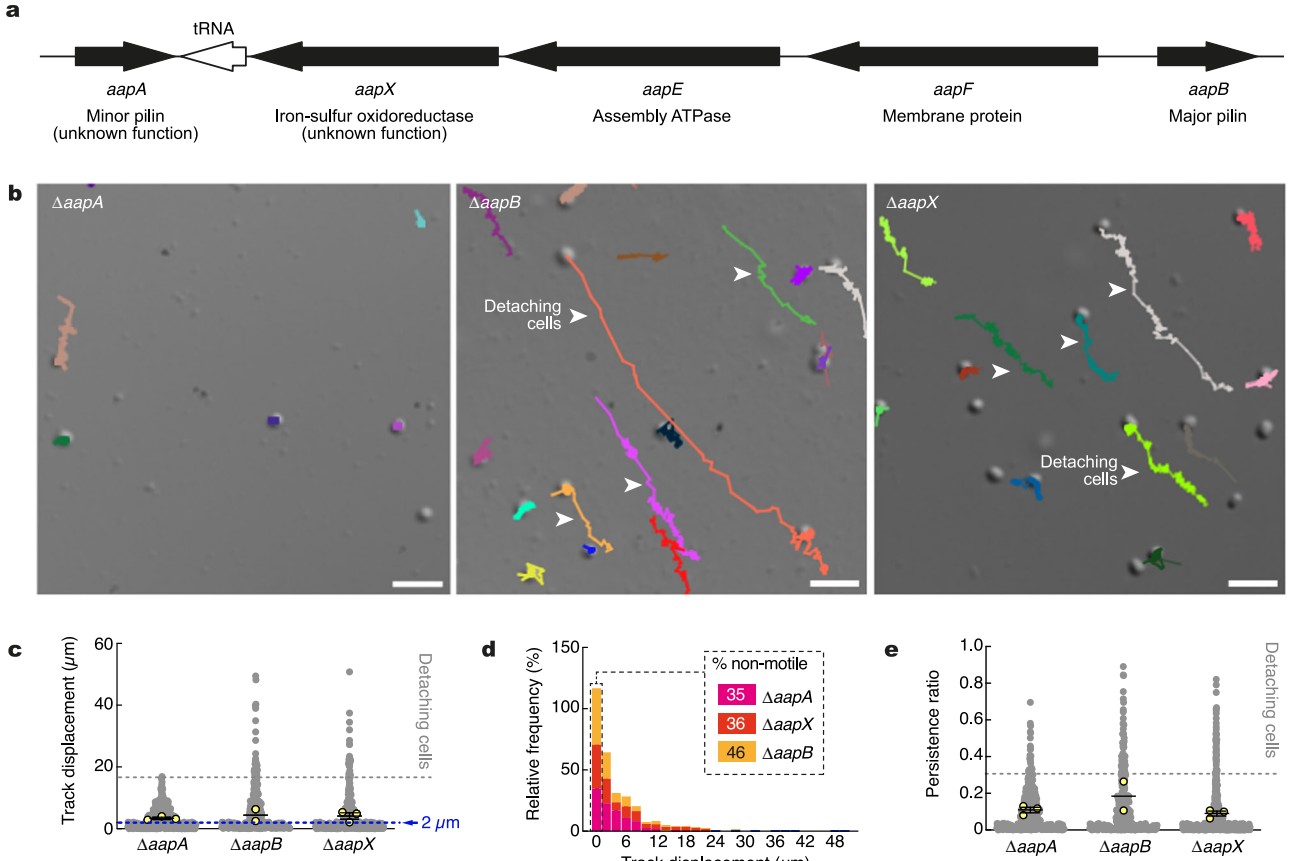

**Fig. 3 | Automated tracking of *S. acidocaldarius* Aap mutants. a** Map of the Aap pilus operon in *S. acidocaldarius*. **b** Representative examples of tracks generated for indicated Aap mutants. Arrowheads denote long aligned tracks generated by detaching cells. Scale bars = 5 μm. **c** Track displacement measured over five minutes for each indicated strain (Δ*aapA*: N = 418, Δ*aapB*: N = 302, Δ*aapX*: N = 427). **d** Distribution of individual track displacements in the indicated strains. Tracks with a displacement <2 μm were defined as non-motile. **e** Track persistence ratio in the indicated strains. Superimposed yellow circles represent sample means from each of n = 3 (Δ*aapA*, Δ*aapX*) or n = 2 (Δ*aapB*) independent experiments. Error bars represent the mean ± SEM of those biological replicates for Δ*aapA* and Δ*aapX*. Source data are provided as a Source Data file.

plasmids in wild-type cells to observe whether Aap pili could still retract in the conditions, but these experiments proved difficult to interpret since very tight inducible promoters do not exist for expression of proteins in *Sulfolobus* species. Although a role for AapE cannot be ruled out, another possibility involves the physical properties of Aap pili. A recent cryo-electron microscopy study of the structure of the Aap pilus proposed that inter-subunit interactions between AapB pilins are compatible with an intrinsic instability[38], similar to *V. cholerae* type IV pili[44].

Recent structural studies by Gaines et al. and Wang et al. suggest that Aap pilus fibers in *Sulfolobus* comprise only the major pilin AapB, and completely lack the minor pilin AapA[13,38]. Cells deficient for the minor pilin AapA, however, are defective in twitching motility, which suggests a possible pilus retraction defect. In bacteria, the function of minor pilins has been a subject of debate, and in some bacterial species, minor pilins appear to exert their function from the cytoplasmic membrane rather than from within the pilus fiber[45]. Similarly, in *Sulfolobus*, AapA might influence pilus assembly and/or retraction from the cytoplasmic membrane. However, we observed pilus retraction in a few motile Δ*aapA* mutant cells, suggesting instead a possible role in pilus-mediated adhesion. Because the structure of the pilus tip was not reconstructed in the recent cryo-electron microscopy studies, we suggest that AapA might be present in minute amounts at the pilus tip, where it mediates substrate adhesion and transduction of mechanical signals. Notably, tip-mediated adhesion has been shown in bacteria[42]. Another possibility is that AapA is present throughout the Aap pilus

fiber in quantities too low to be detected by cryo-electron tomography that uses particle averaging to achieve atomic resolution of protein structures. More structural and biophysical studies will be required to decipher the molecular mechanism of pilus retraction in *Sulfolobus*.

Based on genomic analysis, the retraction ATPase PilT evolved in bacteria as a result of a gene duplication of the assembly ATPase PilB, followed by a separation of the polymerization and retraction activities between the two ATPases[28,29]. In the present study, we show that multiple Sulfolobales species are capable of twitching motility and that *Sa solfataricus* type IV pili are capable of retraction, suggesting that PilT-independent pilus retraction might be common in archaea. We note, however, that there is variability in the twitching behavior of different archaeal species species. We observed less efficient twitching of *Sa. solfataricus* and *S. islandicus* than *S. acidocaldarius* and a previous study detected no twitching motility in the piliated halophile *Haloferax volcanii*[7]. These differences likely reflect adaptation to different lifestyles and specific ecological niches. For instance, *S. acidocaldarius* forms thicker and more complex biofilm structures than other Sulfolobales[18] and preferentially colonizes the rocky periphery of hot springs. The ability to self-assemble at specific loations may depend on robust and efficient Aap pilus-mediated motility[17,46].

Along with driving cell motility, pilus retraction in bacterial species such as *E. coli*, *C. crescentus* and *P. aeruginosa* can be hijacked by bacteriophages, which use it to move into close proximity with the host cell surface[9,47–51]. In archaea too, type IV pili can serve as virus receptors[10–13] and, similar to bacteriophages, archaeal viruses may

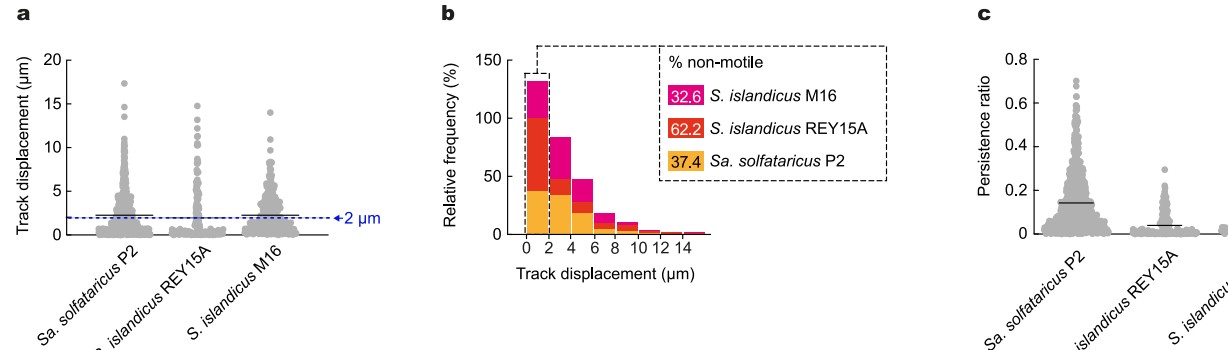

**Fig. 4 | Automated tracking in other Sulfolobales species. a** Track displacement and persistence ratio measured over five minutes for each indicated. **b** Distribution of individual track displacements in the indicated species. Tracks with a displacement under 2 μm correspond to the percentage of non-motile cells. **c** Track persistence ratio in the indicated species. Data represents N = 836 *Saccharolobus solfataricus* P2 cells, N = 180 *Sulfolobus islandicus* REY15A cells and N = 344 *Sulfolobus islandicus* M16.4 cells in n = 1 experiment. Source data are provided as a Source Data file.

ride retracting type IV pili down to the host cell surface to initiate infection.

In summary, the twitching motility we observe in archaeal cells suggests that the ancestral type IV pilus machinery in LUCA was capable of both pilus assembly and retraction, i.e. that retraction is an ancestral trait of type IV pili. This, in turn, implies that LUCA was capable of exploring and colonizing surfaces in the same way as modern prokaryotes.

## Methods
### Cell culture
*Sulfolobus acidocaldarius*, *Sulfolobus islandicus* and *Saccharolobus solfataricus* were grown aerobically at 75 °C with shaking in Brock's minimal media[52]. *S. acidocaldarius* was grown in the presence of 20 μg.mL$^{-1}$ uracil (Sigma-Aldrich).

### Automated cell tracking
*S. acidocaldarius* cells from exponentially growing cultures were imaged in a DeltaT cell micro-environment control system (Bioptechs) at 75 °C as previously described[33] with the difference that the glass coverslip was not modified with solidified media so cells were allowed to adhere to glass, or in a VAHeat chamber controlled by the VAHeat system (Interherence) on a Nikon Ti-E inverted microscope equipped with a 100 × 1.45NA PlanApo objective and a Point Gray CMOS camera (CM3-U3-50S5M-CS FLIR; 85 × 71 μm fields of view), or a Zeiss Observer Z1 microscope equipped with a Plan-Apochromat 100 × 1.40NA objective (166 × 130 μm fields of view). Movies were recorded at a rate of 1 frame per second for a total of 5 min.

The first four frames from a representative movie were used to train the Weka detector[53] in Fiji[54] after decreasing the frame size so that the longest dimension was smaller than 1000 pixels. Weka was trained to detect surface-adhered cells (in focus) and ignore unadhered cells (out of focus) as well as signals from the background, or debris in the culture media. These categories were uploaded in TrakMate 7[55] operated in Fiji and cells were automatically tracked in full-length movies. Low-quality tracks, tracks corresponding to multiple cells wrongfully detected as one, or tracks including cell-cell interaction events were discarded by adjusting the detection threshold in TrackMate 7 and manual pruning. The dataset was refined by visual inspection and manual bridging of gaps within tracks that were interrupted by transient loss of focus or detection defects.

### Mean Square Displacement (MSD) analysis
Mean square displacement (MSD) curves for each track were calculated by averaging the squares of the displacements between all pairs of positions (i.e. at every multiple of the frame rate) within the track.

These MSDs were then plotted as a function of their associated time intervals. Data were shifted to align the starting points and logarithms were computed of the amplitude and time values. Slopes of these individual log-log plots were computed by linear least squares methods.

### Live cell fluorescence imaging of type IV pili dynamics
10 mL of *S. acidocaldarius* cells from an exponentially growing culture were collected by centrifugation at 4000 g at room temperature. The pellet was washed once in phosphate buffered saline (PBS), resuspended in 225 μL PBS, and 25 μL of Alexa Fluor 568-NHS-ester dye (Thermo Scientific) freshly reconstituted at 0.5–1 μg.mL$^{-1}$ in anhydrous DMSO was added. Cells were incubated with the dye for 15 min protected from light, with gentle shaking, at room temperature. Cells were then pelleted, washed twice with minimal Brock's media, and recovered in 5 mL complete Brock's media at 75 °C with shaking for 1.5 h prior to imaging.

Cells were imaged in a VAHeat chamber controlled by the VAHeat system (Interherence) on a Nikon Ti-E inverted microscope equipped with a 100 × 1.45 NA PlanApo objective or a 100 × 1.49 NA Apo TIRF objective, 561 nm laser, Hamamatsu Quest camera, and a VT-iSIM super-resolution module. Movies were recorded at a rate of two frames per second.

### Data analysis
Track analysis and image processing for figure preparation were made in Fiji[54]. Data analysis and preparation of SuperPlots[56] were done in Prism (GraphPad). Figures and illustrations were prepared in Adobe Illustrator.

### Reporting summary
Further information on research design is available in the Nature Portfolio Reporting Summary linked to this article.

## Data availability
None Source data are provided with this paper.

## Code availability
Custom MATLAB code used for MSD analysis is available from the Mullins lab GitHub webpage (https://github.com/mullinslabUCSF/twitchy).

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

## Acknowledgements

This work was supported by the National Institute of General Medical Sciences of the National Institutes of Health (R01-GM061010 to RDM), by the Howard Hughes Medical Institute Investigator Program (RDM), and by a Momentum grant (VW Foundation grant number 94933 to SVA and MVW). ACO is a Simons Fellow of the Life Sciences Research Foundation. The authors thank Rachel Whitaker and Quinxin She for sharing *Sso* and *Sis* strains.

## Author contributions

A.C.O.: conceptualization, methodology, investigation, formal analysis, validation, visualization, figure preparation, writing original draft preparation, and review and editing. M.V.W.: conceptualization, methodology, investigation, formal analysis, validation, visualization, and review and editing. S.L.: methodology, formal analysis, validation, visualization, figure preparation, writing original draft preparation, and review and editing. S.V.A. & R.D.M.: funding acquisition, supervision, methodology, investigation, formal analysis, writing original draft preparation, and review and editing. All authors contributed to the article and approved the submitted version.

## Competing interests

The authors declare no competing interests.
