## [Peer Review File · Nature Communications]

Adhesion pilus retraction powers twitching motility in the thermoacidophilic crenarchaeon *Sulfolobus acidocaldarius*Reviewer #1 (Remarks to the Author):

This manuscript by Charles-Orszag et al. shows that *Sulfolobus acidocaldarius* undergoes a phenomenon that resembles twitching motility in bacteria. They also provide some evidence that this motility is dependent on retraction of Aap pili. The observation that archaeal pili are capable of retraction is very exciting, and the microscopy shown in the manuscript to support this claim is convincing. So, I believe that this study will be an important addition to the field. My enthusiasm was a bit tempered, however, because many of the other statements in the manuscript lack supporting evidence. These points need to be appropriately qualified, or additional data must be provided to support these claims. These are discussed in detail below.

Throughout the study, it is stated that retraction in *Sulfolobus* likely requires a "bifunctional ATPase". While this is a plausible hypothesis, no experimental data is provided to support this statement. Only one bacterial system – *Caulobacter* tad pili - has been shown to require a bifunctional ATPase for retraction. Other bacterial pili that retract in the absence of a dedicated retraction motor ATPase do so through a passive mechanism that relies on the inherent instability of the pilus filament (PMID: 34789573, 30692169). Thus, a distinct possibility is that pilus retraction in *Sulfolobus* is a passive property of the filament. So at the very least, this statement needs to be appropriately qualified in the Abstract and Discussion. Or this should be formally tested experimentally. In *Caulobacter* it was shown that mutations in CpaF that slowed its ATPase activity resulted in an equal slowing of both pilus extension and pilus retraction. Previous work has shown that mutating a conserved position in the walker B motif of PilT and PilB homologs can slow ATPase activity and correspondingly pilus extension/retraction rates (L201C in *Neisseria* PilT, PMID: 27923924; M391C in *Vibrio cholerae* PilB, ref 20; L390C in *Vibrio cholerae* MshE, ref 21). So, similar mutations to aapE could be made to formally test this hypothesis.

Line 59-60. The notion that twitching is non-directional is misleading. Twitching is highly spatially regulated in diverse bacterial species - *Pseudomonas*, *Myxococcus*, and *Cyanobacteria* to name a few – and can promote chemotactic and directed movement. Some citations that support my statement: PMID: 35535498, 34301869, 11133973.

Ref 21 does not show that pili can retract in the absence of a PilT retraction ATPase. Instead, this study demonstrates that the PilT ATPase can also facilitate pilus extension in one T4P system. A more appropriate set of references would be PMID: 34789573 and 30692169, which directly support this point. On a related note, the statement in Lines 241-243 is not accurate. As mentioned above, while there is data to support that the retraction of the *Caulobacter* T4P is dependent on a bifunctional ATPase, in *Vibrio cholerae* it was shown that retraction of the competence T4P and the toxin-coregulated T4P is motor-independent and relies on the inherent instability of the pilus filament (PMID: 34789573).

While Fig 1 shows that Aap pili are likely the structures involved in twitching motility, the labeling in Fig 2 cannot distinguish between pilus structures since it appears that WT cells were used in this experiment. It would be more convincing if the experiments conducted in Fig 2 were also performed on the upsE arlJ double mutant discussed in Fig 1 to eliminate the possibility that the structures being observed in Fig 2 are the UV-inducible pili or the archaellum.

It is unclear why the phenotypes of aapB and aapX differ from the aapF mutant. Especially since all 3 of these mutants should be unpiliated. The representative image for the aapA mutant seems to more closely resemble the aapF mutant. It is difficult to tell, but the quantification may suggest that some of the aap mutants are not distinguishable from WT (compare Fig 1k,m to Fig 3c,e). Although it is difficult to really make this comparison since the data are plotted on separate graphs. At the very least, this needs to be formally compared and commented on. Also, the study heavily relies on the literature to indicate whether or not aap mutants are piliated or not. It is unclear why this is not formally tested using the fluorescent labeling procedure used in Fig 2.

The observation that twitching-like movement is seen in other *Sulfolobales* is compelling. But it is inferred that this movement requires aap-dependent twitching motility. It would be more compelling to actually show that this movement requires pilus retraction by performing experiments similar to those shown in Fig 2.

Reviewer #2 (Remarks to the Author):

Motility plays a crucial role in cellular functions. In bacteria, this movement is facilitated by flagella, while archaea rely on analogous structures called archaella. Bacteria have also been observed to exhibit type IV pili-dependent motility, commonly referred to as twitching motility. Previous research by Orszag et al. demonstrated that *Sulfolobus acidocaldarius*, known for its ability to thrive in low pH and high-temperature environments, also exhibits twitching motility. In the present study, the authors aim to further explore the significance of type IV pili in twitching motility. They employ various techniques such as live-cell imaging, automated cell tracking, and high-temperature fluorescence imaging to visualize the retraction process in wt and numerous mutant strains. Notably, the deletion of *aapF*, encoding a pilus biosynthesis component, results in the loss of twitching motility in cells, making this study a compelling contribution to our understanding of cellular motility mechanisms. However, the authors should address the following concerns.

Major comments:

Title, Line 26-27, etc.:

Please note that the claim of "twitching motility in the absence of a dedicated retraction ATPase" might be inaccurate, as a dedicated retraction ATPase distinct from the known PiIT ATPase, could exist.

Line 28-29:

The authors suggest that their results demonstrate that "the ancestral type IV pilus machinery in the last universal common ancestor (LUCA) relied on such a bifunctional ATPase for both extension and retraction." However, the authors did not previously introduce bifunctional ATPases, and more importantly, the results from the present study do not support this claim.

Line 136:

The data presented strongly suggest Aap involvement. However, the authors should refrain from suggesting that the twitching motility is "specifically" mediated by adhesion pili.

1. Adhesive pili might indirectly affect the functions or expression of other proteins. For example, as mentioned in a previous study and supported in this study, the deletion of *arlJ* increases the expression of Aap pili.

2. In Figure 2, labeling of the surface proteins is nonspecific and was only conducted in the wild type (wt). Therefore, it remains unclear whether the pili shown here are adhesive pili.

3. In Henche et al., 2011, *aapF* was expressed in trans for biofilm formation analysis. The same strain should also be used in this study to confirm the involvement of AapF in twitching motility.

Line 185:

The data presented do not provide evidence that *aapA* is important for retraction.

Figure 2 / movie 2:

Could the authors clarify the frequency of both retraction-mediated cell-cell interactions and the pili dynamics observed here?

Line 243:

Given the absence of supporting data, the authors should consider rephrasing the statement: '...it is likely that the adhesion pili assembly ATPase AapE also coordinates pilus retraction.' Additionally, it would be helpful for the authors to clarify why they opted to show twitching motility phenotypes for the delta *aapF* strain but did not include data for the delta *aapE* strain lacking the assembly ATPase for the adhesion pili. To test the involvement of the UV-inducible pili in twitching motility the authors did chose the strain lacking the *upsE* gene, encoding the assembly ATPase for this pilus.

Minor comments:

Line 18:

Change "substrate adhesion" to "surface adhesion."

Line 22:

"Native conditions" misleading as this only refers to pH and temperature.

Line 38:

The authors mention many functions related to type IV pili but only provide references related to bacteriophages and archaeal viruses.

Line 43:

Please provide a reference.

Line 44:

Change "converted to" to "processed to."

Line 46:

Please provide a reference.

Line 47:

Explain the difference between the assembly platform protein and integral membrane protein and include a reference.

Line 54:

Please replace "in bacteria" as the authors mention in line 64 that some bacteria lack PilT.

Line 65:

Elaborating on PilT-independent twitching motility and bifunctional ATPase here would be helpful.

Line 67:

Change "We observed" to "We previously observed."

Line 68:

Authors have noted that PilT-independent retraction is possible; therefore, the absence of PilT would not necessarily indicate the inability of cells to retract.

Line 73:

Change "The second are the UV-inducible pili (Ups)" to "The second is the UV-inducible pilus (Ups)."

Line 124:

There is inconsistency in the data, with 2% mentioned in the text while 4.2% is shown in Figure 1, panel I.

Line 148:

Please provide a reference.

Figure 3:

Authors should address how Δ apB and Δ apX can still attach to the surface.

Line 200:

Based on the analysis of only four *Sulfolobus* strains, authors should not claim that twitching motility is widespread in archaea.

Charles-Orszag et al. in Nature Communications
Response to reviewers
March 27, 2024

Reviewer #1 (Remarks to the Author)

This manuscript by Charles-Orszag et al. shows that *Sulfolobus acidocaldarius* undergoes a phenomenon that resembles twitching motility in bacteria. They also provide some evidence that this motility is dependent on retraction of Aap pili. The observation that archaeal pili are capable of retraction is very exciting, and the microscopy shown in the manuscript to support this claim is convincing. So, I believe that this study will be an important addition to the field. My enthusiasm was a bit tempered, however, because many of the other statements in the manuscript lack supporting evidence. These points need to be appropriately qualified, or additional data must be provided to support these claims. These are discussed in detail below.

Throughout the study, it is stated that retraction in *Sulfolobus* likely requires a “bifunctional ATPase”. While this is a plausible hypothesis, no experimental data is provided to support this statement. Only one bacterial system – *Caulobacter tad pili* - has been shown to require a bifunctional ATPase for retraction. Other bacterial pili that retract in the absence of a dedicated retraction motor ATPase do so through a passive mechanism that relies on the inherent instability of the pilus filament (PMID: 34789573, 30692169). Thus, a distinct possibility is that pilus retraction in *Sulfolobus* is a passive property of the filament. So at the very least, this statement needs to be appropriately qualified in the Abstract and Discussion. Or this should be formally tested experimentally. In *Caulobacter* it was shown that mutations in CpaF that slowed its ATPase activity resulted in an equal slowing of both pilus extension and pilus retraction. Previous work has shown that mutating a conserved position in the walker B motif of PilT and PilB homologs can slow ATPase activity and correspondingly pilus extension/retraction rates (L201C in *Neisseria* PilT, PMID: 27923924; M391C in *Vibrio cholerae* PilB, ref 20; L390C in *Vibrio cholerae* MshE, ref 21). So, similar mutations to aapE could be made to formally test this hypothesis.

> We thank the reviewer for raising this point and for pointing us to relevant literature. We agree that our hypothesis about the bifunctionality of the assembly ATPase AapE was overstated. We have now softened this claim throughout the manuscript and discussed alternative molecular mechanisms for *Sulfolobus* Aap pilus retraction supported by appropriate references in the discussion section (lines 268-290).

Line 59-60. The notion that twitching is non-directional is misleading. Twitching is highly spatially regulated in diverse bacterial species - *Pseudomonas*, *Myxococcus*, and *Cyanobacteria* to name a few – and can promote chemotactic and directed movement. Some citations that support my statement: PMID: 35535498, 34301869, 11133973.

> Again, we thank the reviewer for helping us increase the accuracy of the paper. This was now included in the introduction section (lines 58-61), and lack of directionality is no longer used as an argument to determine whether *Sulfolobus*

motility is twitching, but as one feature of twitching specific to *Sulfolobus* in the present study (lines 112-118).

Ref 21 does not show that pili can retract in the absence of a PilT retraction ATPase. Instead, this study demonstrates that the PilT ATPase can also facilitate pilus extension in one T4P system. A more appropriate set of references would be PMID: 34789573 and 30692169, which directly support this point. On a related note, the statement in Lines 241-243 is not accurate. As mentioned above, while there is data to support that the retraction of the *Caulobacter* T4P is dependent on a bifunctional ATPase, in *Vibrio cholerae* it was shown that retraction of the competence T4P and the toxin-coregulated T4P is motor-independent and relies on the inherent instability of the pilus filament (PMID: 34789573).

> This has now been corrected.

While Fig 1 shows that Aap pili are likely the structures involved in twitching motility, the labeling in Fig 2 cannot distinguish between pilus structures since it appears that WT cells were used in this experiment. It would be more convincing if the experiments conducted in Fig 2 were also performed on the *upsE arlJ* double mutant discussed in Fig 1 to eliminate the possibility that the structures being observed in Fig 2 are the UV-inducible pili or the archaellum.

> This is a great point, and Fig. 2 now shows data from super-resolution fluorescence high-temperature imaging of pili dynamic performed in the double mutant $\Delta upsE \Delta arlJ$. In this strain, indeed, only adhesion pili can be produced, making very clear the molecular identity of the fibers that we image. Similar to previously presented on the WT strain (which were moved to the Supplementary Figure 2 as a control), adhesion pili are shown to retract and to mediate twitching motility as well as dynamic cell-cell interactions (Fig. 2, Supplementary Videos 2 to 4, lines 149-169). In single retraction events recorded in three independent experiments, adhesion pilus retraction was $\sim 0.3 \mu\text{m}\cdot\text{sec}^{-1}$. We discuss this new value in the discussion section (lines 278-280).

We believe this new data unambiguously demonstrate that twitching motility is specifically mediated by the retraction of adhesion pili. Accordingly, we propose a new title for the manuscript, "*Adhesion pilus retraction powers twitching motility in the thermoacidophilic crenarchaeon *Sulfolobus acidocaldarius", which reflects this main conclusion of the paper and avoids mentioning the absence of PilT.**

It is unclear why the phenotypes of *aapB* and *aapX* differ from the *aapF* mutant. Especially since all 3 of these mutants should be unpiliated. The representative image for the *aapA* mutant seems to more closely resemble the *aapF* mutant. It is difficult to tell, but the quantification may suggest that some of the *aap* mutants are not distinguishable from WT (compare Fig 1k,m to Fig 3c,e). Although it is difficult to really make this comparison since the data are plotted on separate graphs. At the very least, this needs to be formally compared and commented on.

> This is a good point. We believe that $\Delta aapF$ might have a higher level of Aap pilus-independent surface adhesion due to its hyperarchaellation phenotype (described in Henche et al., 2012, PMID 22059595), consistent with some tracks showing a tethered behavior in the new mean square displacement analysis showed in Fig. 1. Neither $\Delta aapB$ or $\Delta aapX$ are known to be hyperarchaellated, hence a looser residual surface adhesion in the absence of Aap pili explaining the long tracks that align with the flow in the chamber (as explained lines 192-194). This is now more clearly discussed (lines 249-255).

Also, the study heavily relies on the literature to indicate whether or not aap mutants are piliated or not. It is unclear why this is not formally tested using the fluorescent labeling procedure used in Fig 2.

> We now confirmed that the $\Delta aapA$ mutant is piliated 1) by super-resolution fluorescence live-cell imaging (Supplementary Figure 2 and lines 197-201) and 2) by transmission electron microscopy (Supplementary Figure 3 and lines 186). We also show that adhesion pili in that mutant can still retract, which is now discussed (lines 292-308).

The observation that twitching-like movement is seen in other Sulfolobales is compelling. But it is inferred that this movement requires aap-dependent twitching motility. It would be more compelling to actually show that this movement requires pilus retraction by performing experiments similar to those shown in Fig 2.

> Type IV pili labeling in other Sulfolobales proved very challenging. While different reactive fluorescent dyes with different chemical properties (molecular weight, solubility) may work, testing other dyes would have significantly delayed our response. However, the labeling strategy worked well enough in *Saccharolobus solfataricus* to allow the visualization of a few events of type IV pilus retraction, motility, and cell-cell interactions in that species that look similar to what we describe in *Sulfolobus acidocaldarius*. This data is now described in lines 212-215 and shown in Supplementary Videos 9 and 10, and we believe that it makes a stronger point that twitching is also mediated by type IV pilus retraction in other Sulfolobales.

Reviewer #2 (Remarks to the Author)

Motility plays a crucial role in cellular functions. In bacteria, this movement is facilitated by flagella, while archaea rely on analogous structures called archaella. Bacteria have also been observed to exhibit type IV pili-dependent motility, commonly referred to as twitching motility. Previous research by Orszag et al. demonstrated that *Sulfolobus acidocaldarius*, known for its ability to thrive in low pH and high-temperature environments, also exhibits twitching motility. In the present study, the authors aim to further explore the significance of type IV pili in twitching motility. They employ various techniques such as live-cell imaging, automated cell tracking, and high-temperature fluorescence imaging to visualize the retraction process in wt and numerous mutant strains. Notably, the deletion of *aapF*, encoding a pilus biosynthesis component, results in the loss of twitching motility in cells, making this study a compelling contribution to our understanding of cellular motility

mechanisms. However, the authors should address the following concerns.

Major comments:

Title, Line 26-27, etc.:

Please note that the claim of “twitching motility in the absence of a dedicated retraction ATPase” might be inaccurate, as a dedicated retraction ATPase distinct from the known PiIT ATPase, could exist.

> We thank the reviewer for their comment, and we agree that our claim may have been overstated. We modified the text accordingly (lines 26-28) and throughout the manuscript. We also propose the new title for the manuscript, “Adhesion pilus retraction powers twitching motility in the thermoacidophilic crenarchaeon *Sulfolobus acidocaldarius*”, which reflects the main conclusion of the paper and avoids mentioning the absence of PiIT.

Line 28-29:

The authors suggest that their results demonstrate that “the ancestral type IV pilus machinery in the last universal common ancestor (LUCA) relied on such a bifunctional ATPase for both extension and retraction.” However, the authors did not previously introduce bifunctional ATPases, and more importantly, the results from the present study do not support this claim.

> In line with the previous comment, we agree that this may have been overstated. We have now discussed alternative molecular mechanisms for pilus retraction and modified the text accordingly throughout the manuscript (lines 268-290).

Line 136:

The data presented strongly suggest Aap involvement. However, the authors should refrain from suggesting that the twitching motility is “specifically” mediated by adhesion pili.

1. Adhesive pili might indirectly affect the functions or expression of other proteins. For example, as mentioned in a previous study and supported in this study, the deletion of *arlJ* increases the expression of Aap pili.

2. In Figure 2, labeling of the surface proteins is nonspecific and was only conducted in the wild type (wt). Therefore, it remains unclear whether the pili shown here are adhesive pili.

3. In Henche et al., 2011, *aapF* was expressed in trans for biofilm formation analysis. The same strain should also be used in this study to confirm the involvement of AapF in twitching motility.

> This is a good point, and Fig. 2 now shows data from super-resolution fluorescence high-temperature imaging of pili dynamic performed in the double mutant $\Delta upsE\Delta arlJ$. In this strain, indeed, only adhesion pili can be produced, making very clear the molecular identity of the fibers that we image. Similar to previously presented on the WT strain (which were moved to the Supplementary Figure 2 as a control), adhesion pili are shown to retract and to mediate twitching motility as well as dynamic cell-cell interactions (Fig. 2, Supplementary Videos 2 to 4, lines 149-169). In single retraction events recorded in three independent experiments, adhesion

pilus retraction was $\sim 0.3 \mu\text{m}\cdot\text{sec}^{-1}$. We discuss this new value in the discussion section (lines 277-279).

Line 185:

The data presented do not provide evidence that *aapA* is important for retraction.

> We now show by super-resolution fluorescence live-cell imaging (Supplementary Figure 2 and lines 197-199) that adhesion pili in that mutant can still retract, which is now discussed (lines 292-308). We also confirm that the $\Delta aapA$ mutant is piliated by transmission electron microscopy (Supplementary Figure 3).

Figure 2 / movie 2:

Could the authors clarify the frequency of both retraction-mediated cell-cell interactions and the pili dynamics observed here?

> Aap retraction-mediated dynamic cell-cell interactions were readily observed in WT and $\Delta upsE\Delta arlJ$. However, these events depend on a combination of (at least) local cell density, number of Aap pili per cell, and Aap pilus length. As a result, it is challenging (if not impossible) to provide a frequency at which these events happen. A more controlled assay could be sought of in future studies, but we believe this would be out of the scope of the present work.

Line 243:

Given the absence of supporting data, the authors should consider rephrasing the statement: '...it is likely that the adhesion pili assembly ATPase *AapE* also coordinates pilus retraction.' Additionally, it would be helpful for the authors to clarify why they opted to show twitching motility phenotypes for the $\Delta aapF$ strain but did not include data for the $\Delta aapE$ strain lacking the assembly ATPase for the adhesion pili. To test the involvement of the UV-inducible pili in twitching motility the authors did chose the strain lacking the *upsE* gene, encoding the assembly ATPase for this pilus.

> We agree with the reviewer, and we now discuss alternative mechanisms for PilT-independent pilus retraction (lines 268-290). As for why we did not use an *aapE* mutant strain, it is for technical reasons. As opposed to *upsE*, not all assembly ATPases have been amenable to mutagenesis in the past, and *aapF* (for Aap pili) and *arlJ* (for archaealla) were used instead and proved to disrupt the assembly of specific type IV pili (Henche et al., 2012; PMID 22059595).

Minor comments:

Line 18:

Change "substrate adhesion" to "surface adhesion."

> Edited

Line 22:

"Native conditions" misleading as this only refers to pH and temperature.

> Changed to "physiologically relevant"

Line 38:

The authors mention many functions related to type IV pili but only provide references related to bacteriophages and archaeal viruses.

> Relevant references are now added

Line 43:

Please provide a reference.

> Done

Line 44:

Change "converted to" to "processed to."

> Changed

Line 46:

Please provide a reference.

> Added

Line 47:

Explain the difference between the assembly platform protein and integral membrane protein and include a reference.

> The mistake was corrected and a reference added

Line 54:

Please replace "in bacteria" as the authors mention in line 64 that some bacteria lack PiIT.

> Replaced by "in most bacterial type IV pili", as the absence of PiIT is an exception

Line 65:

Elaborating on PiIT-independent twitching motility and bifunctional ATPase here would be helpful.

> This is now discussed in more detail

Line 67:

Change "We observed" to "We previously observed."

> Corrected

Line 68:

Authors have noted that PiIT-independent retraction is possible; therefore, the absence of PiIT would not necessarily indicate the inability of cells to retract.

> The absence of PiIT is an exception in the bacterial world, and it was assumed that archaeal type IV pili were unable to retract (PMID 19347566)

Line 73:

Change "The second are the UV-inducible pili (Ups)" to "The second is the UV-inducible pilus (Ups)."

> Modified

Line 124:

There is inconsistency in the data, with 2% mentioned in the text while 4.2% is shown in Figure 1, panel I.

> Thank you, this has now been corrected

Line 148:

Please provide a reference.

> Added

Figure 3:

Authors should address how Δ aapB and Δ aapX can still attach to the surface.

> This point is now discussed (lines 249-255)

Line 200:

Based on the analysis of only four Sulfolobus strains, authors should not claim that twitching motility is widespread in archaea.

> This claim has been softened.

Reviewer #1 (Remarks to the Author):

The authors have addressed all of the points I raised in the first round of review. I don't have any further comments and believe that this manuscript provides a very important advance to our understanding of archeal type IV pili.